# Enhanced dietary reconstruction of Korean prehistoric populations by combining $\delta^{13}C$ and $\delta^{15}N$ amino acids of bone collagen

**Kyungcheol Choy** [1]*, **Hee Young Yun**[2], **Benjamin T. Fuller**[3], **Marcello A. Mannino**[3]

**1** Department of Cultural Anthropology, Hanyang University ERICA, Ansan, South Korea, **2** Department of Marine Sciences and Convergence Engineering, Hanyang University ERICA, Ansan, South Korea, **3** Department of Archaeology and Heritage Studies, School of Culture and Society, Aarhus University, Højbjerg, Denmark

* kchoy@hanyang.ac.kr

**Data Availability Statement:** All relevant data are within the paper and its Supporting information files.

## Abstract

Compound specific stable isotope analysis of amino acids (CSIA-AA) is a powerful tool for determining dietary behaviors in complex environments and improving dietary reconstructions. Here, we conducted CSIA-AA on human (n = 32) and animal (n = 13) remains from two prehistoric archaeological sites (Mumun, Imdang) to assess in more detail the dietary sources consumed by prehistoric Korean populations. Results of estimated trophic position (TP) using $\Delta^{15}N_{Glx-Phe}$ show that the Imdang individuals consumed aquatic resources, as well as terrestrial resources. Principal component analysis (PCA) using $\delta^{13}C$ and $\delta^{15}N$ essential amino acid (EAA) values show that the Imdang humans closely cluster with game birds and terrestrial herbivores, whilst the Mumun humans closely cluster with $C_4$ plants. Quantitative estimation by a Bayesian mixing model (MixSIAR) indicates that the Imdang humans derived a large proportion of their proteins from terrestrial animals and marine fish, whereas the main protein sources for the Mumun humans were $C_4$ plants and terrestrial animals. Additionally, the comparison between the EAA and bulk isotope models shows that there is a tendency to overestimate the consumption of plant proteins when using bulk isotopic data. Our CSIA-AA approach reveals that in prehistoric Korea there were clear differences in human diets through time. This study adds to a growing body of literature that demonstrates the potential of CSIA-AA to provide more accurate estimations of protein consumption in mixed diets than previous bulk isotopic studies.

## Introduction

In the last decades, carbon and nitrogen stable isotope ratio measurements of bulk collagen have become a common method to investigate dietary patterns and subsistence activities in ancient populations [1–3]. Nevertheless, dietary reconstructions based on bulk stable isotope ratios can be hampered by ecological and metabolic factors [4–6]. For example, numerous food sources have similar bulk $\delta^{13}C$ and $\delta^{15}N$ values, which reduces the ability to fully separate dietary sources [4–7]. Therefore, if humans and animals consumed complex diets, bulk stable

**Funding:** This work was supported by Research Funds for Young Faculty Club at Hanyang University (HY-2018-00000001315) and by the National Research Foundation of Korea (NRF) funded by the Ministry of Education NRF-2022S1A5A2A03051382) and also supported by the Aarhus University Research Foundation (AUFF-E-2015-FLS-8-2).

**Competing interests:** The authors have declared that no competing interests exist.

isotope compositions may not be sufficient to interpret the exact mix of foodstuffs consumed. Moreover, the same plant species at the base of the terrestrial food webs can have a large variability in bulk stable isotope values, owing to the isotopic compositions of their soils of origin and due to possible manuring [8, 9]. Thus, bulk isotopic values of each food source in human diets can vary according to local environments and soil conditions [4, 8, 9]. Furthermore, bulk isotope data can be strongly influenced by metabolic parameters such as dietary routing, and consumer-to-diet isotopic offsets [10]. However, these diverse factors are not often considered in bulk stable isotopic studies. Thus, it can be challenging to determine the isotopic baseline of ancient food webs and discriminate food sources with sufficient accuracy.

In response, new isotopic methods have emerged to reconstruct subsistence practices in past populations [4, 11, 12]. In particular, there has been an increase in the use of compound specific isotope analysis of individual amino acids (CSIA-AA) on bone collagen from archaeological contexts [12–16]. CSIA-AA is a relatively new method that allows more accurate dietary reconstructions in past populations than the bulk isotopic method and consequently provides trophic web reconstructions of past environments [11, 12, 17–19]. Amino acids (AAs) can be divided into two groups on the basis of their synthesis and function: essential amino acid (EAA) and non-essential amino acid (NEAA). While this constitutes the conventional definition of essential (EAA) and nonessential amino acids (NEAA), and this definition is not solely confined to dietary consideration but is also extended to include the role of AAs in supporting protein deposition and growth during different stages of life [20]. The carbon isotope values in AA metabolism are neatly aligned with the conventional distinction between EAA and NEAA. Thus, EAA $\delta^{13}$C values can yield more detailed dietary information since they are directly routed from the diet to the consumer's tissues [21–23]. Therefore, the EAA $\delta^{13}$C values in bone collagen provide detailed information on the carbon sources of the foods consumed, and this can be a useful approach to investigate dietary sources in prehistoric populations [5, 11–13].

In contrast to $\delta^{13}$C values in AA pathways, $\delta^{15}$N AA values do not conform to the conventional definition of EAA and NEAA. Instead, there has been the development of the novel definition of source (Tyr, Lys, Phe, Ser, Gly), metabolic (Thr) and trophic AAs (Pro, Ala, Glx, Val, Leu, Asx) in nitrogen AAs [24–28]. However, it is important to be cautious when applying the meaning of source and trophic AAs in nitrogen to dietary studies as these definitions can be variable due to the species studied and developmental age. For instance, among the known source AAs (Tyr, Lys, Phe, Ser, Gly) in nitrogen, glycine and serine have been found to have a diet-to-consumer offset in pigs ($\Delta^{15}$N $_{Ser}$: 3.2‰ in $C_3$ diets and 2.6‰ in $C_4$ diets, $\Delta^{15}$N$_{Gly}$: 0.8‰ in $C_3$ diets and 2.2‰ in $C_4$ diets) [29]. Only phenylalanine (Phe) appears to exhibit a relative ~0‰ diet-to-consumer offset in many species including humans [26, 30, 31]. Thus, we cannot directly substitute source AAs to EAAs for all animal species and humans, and more research on the definition of source, trophic and metabolic AAs is necessary. However, previous studies have shown that the two $\delta^{15}$N$_{AA}$ values (Glx, Phe) in bone collagen are useful for estimating the trophic level of humans and animals in food webs [15, 18, 31, 32]. The $\delta^{15}$N values of glutamic acid/glutamine (Glx) (trophic AA) become systematically enriched from food sources to consumers, while Phe (source AA) remain relatively unchanged through this process. A comparison of trophic AAs and source AAs ($\delta^{15}$N$_{Glx-Phe}$) can provide a more definitive indicator of trophic position of humans and animals than bulk stable isotope data [33, 34].

Despite the promise of CSIA-AA, most dietary reconstruction research has focused on only one of these measurements, either $\delta^{13}$C$_{AA}$ or $\delta^{15}$N$_{AA}$ values of bone collagen due to the complexity of the methodological development and the relatively high cost in terms of time and analytical expenses. Until now, there have been only a few applications of CSIA-AA using both $\delta^{13}$C$_{AA}$ and $\delta^{15}$N$_{AA}$ of bone collagen from archaeological sites [14–16, 35]. However, the

application of both $\delta^{13}C_{AA}$ and $\delta^{15}N_{AA}$ for dietary reconstructions has increased the possibility of dealing successfully with the issues of food source equifinality, influence of baseline variation, and uncertainties associated with consumer-to-diet isotopic offsets [4]. Thus, it is required to have more applications of both $\delta^{13}C_{AA}$ and $\delta^{15}N_{AA}$ to dietary reconstructions in prehistoric populations from different regions.

In South Korea, the application of bulk isotopic analysis to human remains has recently increased, and covered topics such as crop cultivation, animal domestication, and social status [36–39]. However, there has been only a few studies on prehistoric Korean populations using $\delta^{13}C_{AA}$ [6, 36], but no studies using both $\delta^{13}C_{AA}$ and $\delta^{15}N_{AA}$. In this study, we applied CSIA-AA (both $\delta^{13}C_{AA}$ and $\delta^{15}N_{AA}$) to humans (n = 32) and animals (n = 13) from two Korean prehistoric sites: Mumun and Imdang. The Mumun sites belong to the Bronze Age (1500–100 BC) when early agriculture was first introduced to the Korean peninsula, and the Imdang site is dated to the Proto-Three Kingdom Period (BC 80–394 AD) when the early ancient states emerged on the Korean peninsula. Thus, the dietary practices from the two different-era sites can offer important clues to the spread of early agriculture and the subsistence economy of ancient states in prehistoric Korea. Previous bulk isotope data showed that $C_4$ plants such as millets were the main dietary sources in the Mumun sites [36], and there was no consumption of aquatic resources in the diet of the Mumun. On the other hand, at the Imdang site, $C_3$ plants and terrestrial animals were the main dietary sources [37], and bulk isotope studies suggest that notable proportions of marine animal protein were consumed. However, with bulk stable isotope data, it is difficult to identify minor resources in the millet-dominated Mumun diets or to separate each source in the multiple food groups in the Imdang diets.

Along with the CSIA-AA, we also employ a Bayesian mixing model to estimate the proportional contribution of food sources for each individual. In archaeology, there are two widely used Bayesian models: FRUITS [40] and MixSIAR [41, 42]. The MixSIAR can be used to estimate the contribution of diets in ecological, as well as archaeological studies. In this study, we used the MixSIAR to compare the estimated dietary compositions of two or more groups, provided all groups are directly comparable [10, 42]. For the dietary sources in the model, we additionally measured both $\delta^{13}C_{AA}$ and $\delta^{15}N_{AA}$ of animal dietary sources (marine animals, game birds, terrestrial herbivores) from the archaeological sites, and plant sources (n = 9) from modern farms in South Korea. Overall, the focus of this study is to reveal the significance of the domesticated grains (rice *vs*. millet) in Mumun diets, and to more accurately distinguish the dietary sources of the two above-mentioned prehistoric Korean populations using the CSIA-AA (both $\delta^{13}C_{AA}$ and $\delta^{15}N_{AA}$), and to quantify the dietary variability in more detail, using MixSIAR.

## Archaeological context

**Mumun burials.**   The three Mumun burial sites (Hwangsok-ri, Jungdo, Maedun) are all located around the upstream areas of the Han River on the eastern Korean Peninsula [36] (Fig 1). The radiocarbon dates on the human collagen from the three Mumun burials showed that they belong to the Middle Mumun (535–635 BC) period [36]. Korean archaeologists refer to the Mumun period (1500–100 BC) as the Bronze Age in Korean prehistory. The Mumun culture is characterized by crop agriculture with large dry-fields, and settlements with ditch enclosures [43]. Most of the Mumun sites in Korean prehistory are related to human burials, such as cemeteries that do not contain animal remains. Thus, there are very few animal remains recovered from Mumun sites. However, it is suggested that people in the Mumun period continued to fish and hunt wild animals. Archaeobotanical data show that the Mumun people cultivated rice, which was believed to be the most important crop [44–46]. It is assumed that the

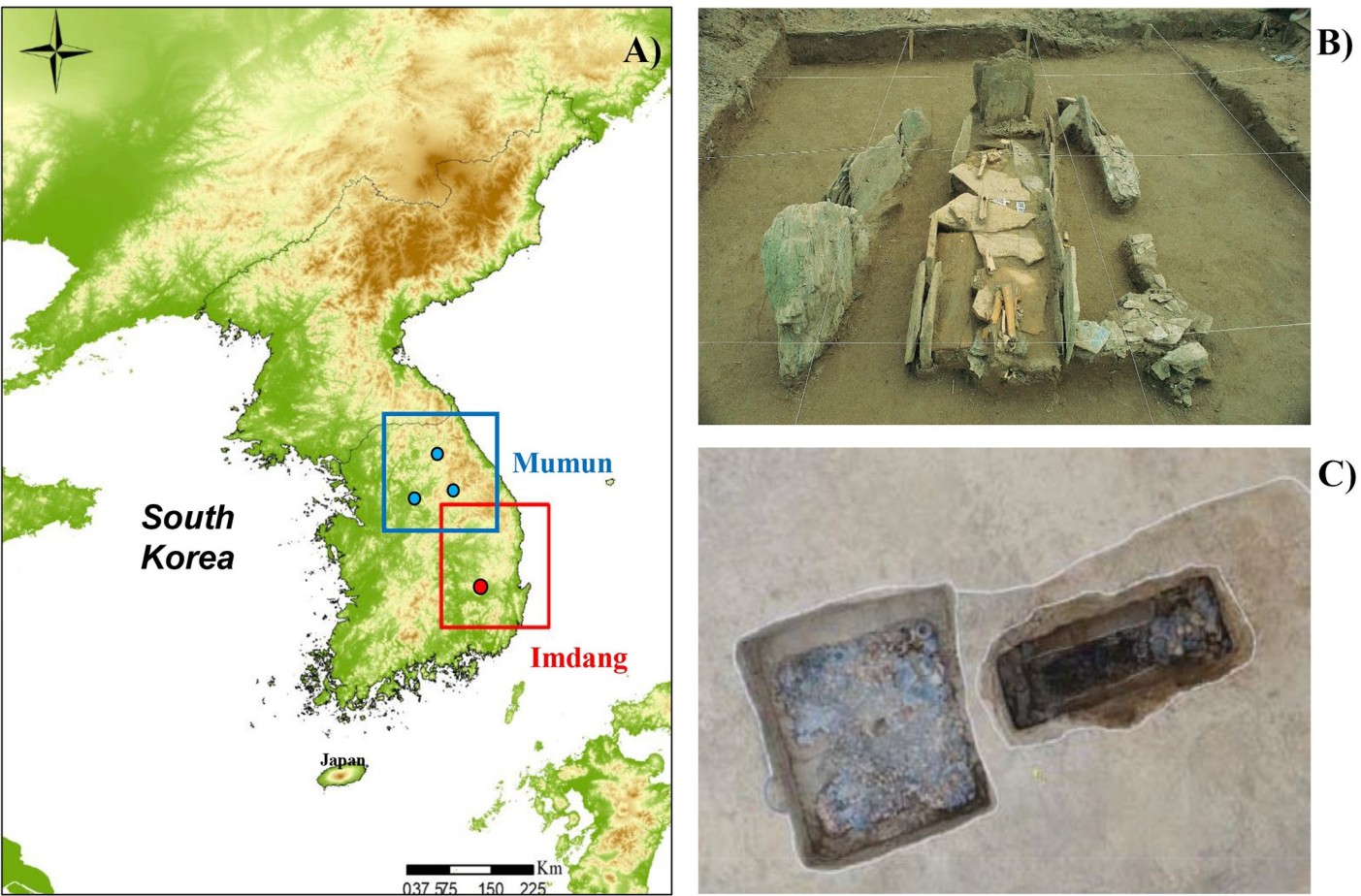

**Fig 1.** Map of the Korean Peninsula showing the location of the Mumun and Imdang sites studied in this paper (A), and photos showing the two types of burials from the different time periods: exposure of the Mumun burial (*Goindol*) from Hwangsuk-ri after removing the cover stone (B) and the Imdang burial composed of a double chamber (square and rectangle chamber) (C).

social and economic complexities during the Mumun period are associated with the spread of rice agriculture. In addition to rice, the Mumun also cultivated millets, barley, wheat, legumes [44, 47]. In particular, studies on charred grains have suggested that millets might also be a dominant crop in the Mumun diets along with rice [44]. However, despite abundant evidence for the cultivation of millets during the Mumun period, it has been difficult to establish the significance of the domesticated grains (rice *vs*. millet) in Mumun diets.

 **Imdang burial mounds.** The Imdang burial mounds constitute a well-known archaeological site in South Korea, located on a hillside in the middle of an alluvial plain that was formed by the Kumho River near Gyeongsan City [48] (Fig 1). This burial site contained approximately 1600 burials and a large amount of grave goods such as gilt-bronze crowns and ornaments, pottery, iron weapons and tools, as well as human and animal bones [49, 50]. Radiocarbon dates confirm that this mortuary site was occupied during the Proto-Three Kingdoms period (BC 80–394 AD) [37]. In Korean archaeology, the Proto-Three Kingdom Period is characterized by the occurrence of local polities with intensive agriculture, an increase in iron production, and new pottery production techniques [51, 52]. Archaeobotanical data show that plant resources in Imdang society include domesticated crops (rice, foxtail millet,

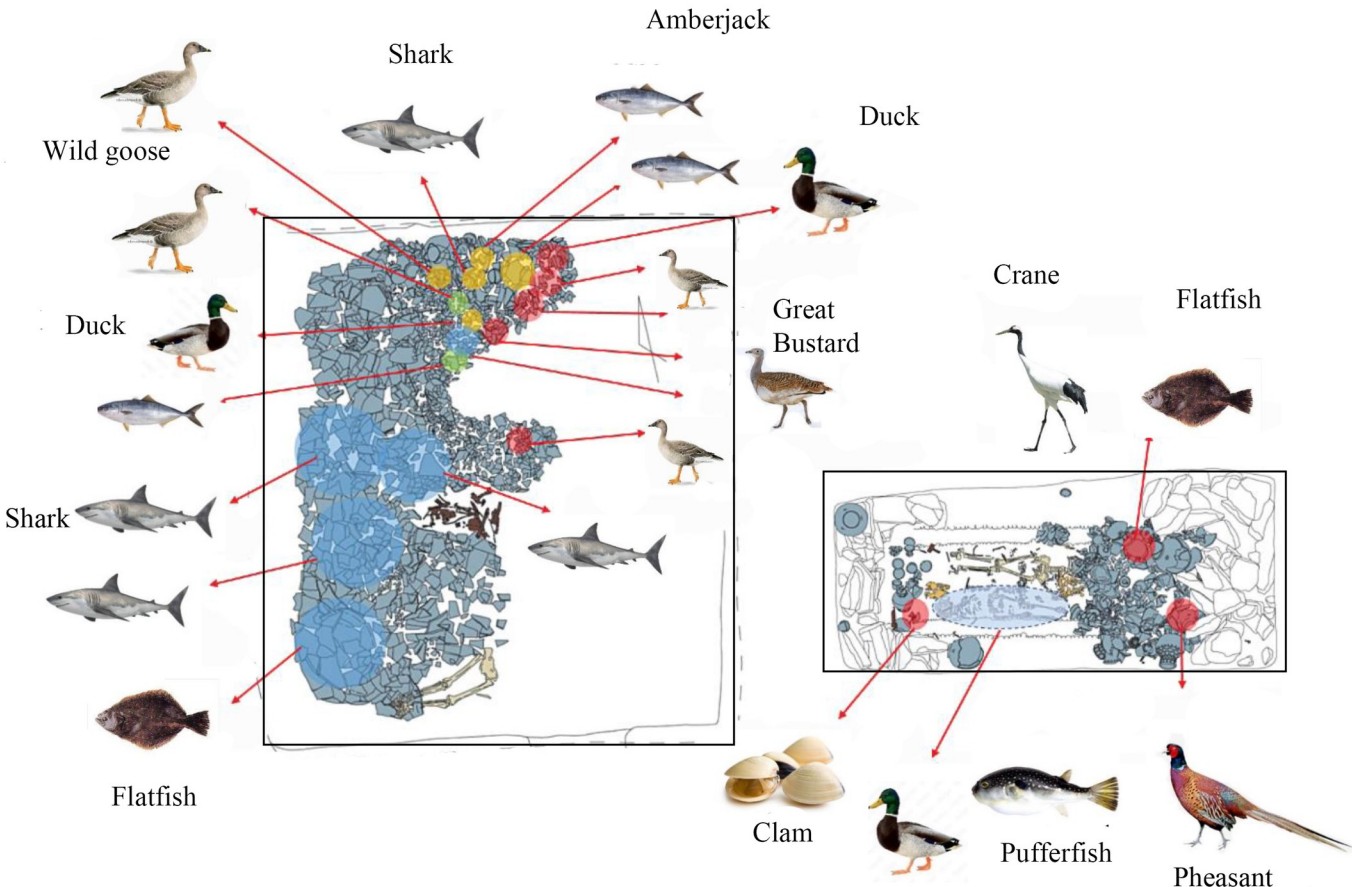

**Fig 2. Example of animal species found in an Imdang burial (Joyeung EII-2ho).** The Imdang burial is composed of a double chamber tomb, with a main chamber (rectangle) and auxiliary chamber (square). Each chamber contains sacrificed humans (*sunjang*) and a variety of wild birds (pheasant, wild goose, great bustard, duck, crane) and marine fish (shark, amberjack, sea bream, flatfish, puffer fish, shellfish), which were used as ritual offerings in the owner's funeral.

barnyard millet, perilla) and fruits (peach, persimmon, apricot). In particular, rice grains were found on the floor of the chamber and inside pottery [53]. Further, the Imdang burials also contained the skeletal remains of terrestrial and marine animals. Animal species found at the Imdang site include game birds (pheasant, wild goose, great bustard, swan), terrestrial herbivores (deer, wild boar, hare), and marine fauna (shark, amberjack, sea bream, rockfish, puffer fish, shellfish) [53, 54] (Fig 2). According to the zooarchaeological studies, the most represented terrestrial animals were game birds such as pheasant and wild goose [53–55] and the most common marine animals were shark, amberjack and shellfish [54, 56].

## Materials and methods

### Archaeological humans and animals

We selected well-preserved humans (n = 4) and animals (n = 2) from the Mumun sites and humans (n = 28), and animals (n = 11) from the Imdang site. It should be noted that it is extremely difficult to obtain humans and animals from the Mumun period as these remains are recovered from inland mountain areas of South Korea that are not good for bone preservation due to acidic soil conditions. Thus, this is the reason why so few specimens from the

Mumun sites were analyzed in this study. For the Mumun sites, the faunal specimens selected are one deer and wild boar, which were analyzed to provide $\delta^{13}C_{AA}$ and $\delta^{15}N_{AA}$ values of terrestrial sources in the archaeological context. However, numerous well-preserved human and animal specimens were recovered from the Imdang mounds dated to the Proto-Three Kingdom period. We selected Imdang humans (n = 28) to investigate dietary patterns among individuals. As potential dietary sources at the Imdang site, we chose pheasant (n = 1), great bustard (n = 1), wild goose (n = 1) and swan (n = 1) for game birds, and we selected hare (n = 1), wild boar (n = 1), pig (n = 1) and cattle (n = 1) for terrestrial animals. In terms of marine species, sandbar shark (n = 2) and amberjack fish (n = 1) were chosen for the CSIA-AA (both $\delta^{13}C_{AA}$ and $\delta^{15}N_{AA}$). These marine and terrestrial animals in the Imdang burials were selected as likely food sources for the CSIA-AA (Fig 2).

## Modern cereals

In order to reconstruct the human diets in prehistoric Korea, it is necessary to have isotopic data of ancient plants (grains, fruits, and vegetables). However, no plant samples from the two prehistoric Korean sites were available to the CSIA-AA due to the inadequate preservation of floral remains, and there is no published isotopic data on plant remains from the Mumun and Proto-Three Kingdom periods. Thus, we collected $C_3$ crops (rice, soybean, adzuki bean, wheat, barley, and oat) (n = 6) and $C_4$ crops (common millet, fox millet, and sorghum) (n = 3) from modern farms in South Korea. These edible cereals are commonly found in archaeological sites dated to the Mumun and Proto-Three Kingdom periods [57], and they were measured for the CSIA-AA as reference materials for potential plant sources in prehistoric Korea.

## Collagen extraction

Collagen extraction was completed at the Moesgaard Archaeo-Science Laboratory (Department of Archaeology and Heritage Studies, Aarhus University) according to the protocols outlined in Richards and Hedges [58] with the addition of an ultrafiltration step [59]. Bone samples were cleaned and demineralized in 0.5M HCl at 4˚C. Demineralized bones were then rinsed in deionized water and gelatinzed at 70˚C in a pH 3 solution for 48 hours. The insoluble fraction was then filtered, first with 5mm Ezee filters and then the remaining solution was filtered to collect purified collagen using >30 kDa filters. The purified solution was then frozen and freeze- dried for 48 hours.

## Preparation of amino acids for GC-C-IRMS

For AA hydrolysis, 1 mg of extracted bone collagen was hydrolyzed with 1 mL 6M HCl (110˚C, 24 h). Modern cereal samples were homogeneously powdered, washed three times with deionized water, and freeze-dried. Lipids were first extracted from the powdered cereal samples with dichloromethane/methanol (2:1 v/v, 10ml) by ultra-sonication. Approximately, 3 mg of dried powders were hydrolyzed (6M HCl, 2ml, 110˚C, 24 h). The AA hydrolysates for the cereal samples were separated from the lipophilic fraction by adding n-hexane/dichloromethane (6:5, v/v, 2mL). A standard mixture of 12 AAs was prepared alongside the samples (Sigma-Aldrich Chemie; Fluka Chemie). An internal standard (14 μl of norleucine, Sigma-Aldrich) was added, and the hydrolyzed collagen and cereal samples were blown down to dryness under $N_2$ and prepared for AA derivatization [30–32].

For $\delta^{15}N_{AA}$ measurements, AAs were derivatized to using the N-pivaloyl-i-isopropyl ester (NPIP) method [60]. For preparation of the NPIP derivatives, the hydrolysate was first esterified using a thionyl chloride/2-propanol (v/v, 1:4, 110˚C, 2h) and then derivatized using pivaloyl chloride/dichloromethane (v/v, 1:4, 110˚C, 2h). Subsequently, the AA derivatives were

stored in 250 μL dichloromethane at -20˚C in a freezer until analysis. For $\delta^{13}C_{AA}$ measurements, a different AA derivatization procedure using N-acetyl methyl ester (NACME) method was employed. [60]. For the preparation of the NACME derivatives, the hydrolysates were methylated by acidified methanol (v/v, 1:4, 75˚C, 1h). Each sample was acetylated with acetic anhydride/triethylamine/acetone (v: v: v, 1:2:6) for 10 min at 60˚C. Following each reaction, reagents were evaporated under nitrogen. The AA derivatives were stored in ethyl acetate (250 μL) at -20˚C in a freezer until GC-C-IRMS analyses.

## Gas chromatography-combustion-isotope ratio mass spectrometry (GC-C-IRMS)

Gas chromatography-combustion-isotope ratio mass spectrometry (GC-C-IRMS) was performed at the Department of Marine Sciences, Hanyang University, South Korea. For $\delta^{15}N_{AA}$ results, these were analyzed with a gas chromatograph (HP 6890N, Agilent Technologies, US) connected to a furnace (GC5 Interface, Elementar, Germany) and an isotope ratio mass spectrometer (Isoprime 100, Elementar, Germany). HP-Ultra 2 (50 m length, 0.32 mm inner diameter, 0.52 μm film thickness) was equipped to the GC. The initial oven temperature was 40˚C for first 5 min. The oven temperature was increased at a rate of 15˚C/min to 110˚C, a rate of 3˚C/min to 150˚C, a rate of 6˚C/min to 220˚C (10 min hold), and a rate of 20˚C/min to 250˚C (8 min hold). As the NPIP derivatization procedure can potentially result in differences between the proline and hydroxyproline [61], the collagen specimens were again derivatized with the NACME protocol to confirm the proline and hydroxyproline $\delta^{15}N$ results. Samples were injected into the GC, equipped with VF-23ms column (30 m length, 0.32 mm inner diameter, 0.25 μm film thickness). The initial oven temperature was 60˚C for the first 1 min. The oven temperature was increased at a rate of 15˚C/min to 120˚C, a rate of 3˚C/min to 190˚C, and a rate of 15˚C/min to 250˚C (15 min hold). The initial oven temperature was 60˚C for the first 1 min. The oven temperature was increased at a rate of 15˚C/min to 120˚C, a rate of 3˚C/min to 190˚C, and a rate of 15˚C/min to 250˚C (15 min hold). 10 certified AA standards were used, which were purchased from SHOKO-Science (Japan) and Indiana University (USA). In addition, a standard mixture of 12 AAs (alanine, glycine, valine, leucine, isoleucine, proline, aspartic acid, threonine, serine, glutamic acid, phenylalanine, and hydroxyproline) was run for testing instrumental and analytical accuracy. Standard nitrogen isotope ratios were expressed as $\delta^{15}N$ value per mil relative to Ambient Inhalable Reservoir (AIR). Analytical error (SE) of the $\delta^{15}N_{AA}$ measurement of the standard AAs was checked, and ranged from 0.31‰ for $\delta^{15}N$ glycine to 0.79‰ for $\delta^{15}N$ threonine.

The $\delta^{13}C_{AA}$ were analyzed with an IRMS instrument (visION, Isoprime) connected with a Hewlett Packard 7890 N series gas chromatograph (Agilent Technologies) via a combustion interface (glass tube packed with copper oxide, operated at 850˚C). Samples were injected into the GC, equipped with VF-23ms column (30 m length, 0.32 mm inner diameter, 0.25 μm film thickness) (at a constant 1.3 mL/min flow rate) as a carrier gas. The initial oven temperature was 60˚C for first 1 min. The oven temperature was increased at a rate of 15˚C/min to 120˚C, a rate of 3˚C/min to 190˚C, and a rate of 15˚C/min to 250˚C (15 min hold). Standard carbon isotope ratios were expressed as $\delta^{13}C$ value per mil relative to Vienna-PeeDee Belemnite (VPDB). Standard deviations of carbon isotope measurements were generally ±0.4‰ as determined by repeated injections of the 10 AA standard mixture and each measured mean AA $\delta^{13}C$ was corrected relative to the mean AA $\delta^{13}C$ of standards to account for the exogenous carbon and kinetic fractionation introduced during derivatization [62]. The $\delta^{13}C$ values of the underivatized AAs in the standard were measured via EA-IRMS, and were used in combination with the measured $\delta^{13}C$ values of their derivatives to calculate correction factors for each

amino acid, as described in Silfer et al. [63]. These correction factors were then used to calculate the $\delta^{13}C$ values of the AAs in the samples from the $\delta^{13}C$ values of their derivatives. Analytical error (SE) of $\delta^{13}C_{AA}$ measurements was estimated from the AA standards, and ranged from 0.10‰ for $\delta^{13}C$ valine to 0.45‰ for $\delta^{13}C$ phenylalanine.

## Statistical analysis

Prior to statistical analysis, all isotopic data were tested for univariate normality by Q-Q plots. According to ecological, archaeological and isotopic relevance, potential food sources were categorized into five food groups: terrestrial herbivores (i.e., deer, wild boar, hare), marine fish (i.e., sandbar shark, amberjack), game birds (i.e., pheasant, bustard, wild goose, and swan), $C_3$ plants (i.e., rice, wheat, barley, oat, adzuki bean, soybean), and $C_4$ plants (foxtail millet, broom-corn millet, sorghum). For example, wild goose and swan as waterfowl have different habitats from terrestrial birds, such as pheasant and the great bustard in terms of ecological niche. However, due to their terrestrial-based isotopic similarity, we grouped these waterfowl as game birds [37]. Isotopic data on three animal groups (game birds, marine fish, terrestrial herbivores) from the Imdang burials, and two plant groups ($C_3$, $C_4$ plants) from modern farms were used for the comparison of dietary sources. For the separation of dietary sources, we used principal component analysis (PCA) to compare EAA $\delta^{13}C$ and $\delta^{15}N$ values of human collagen in relation to the five food groups. EAAs are those that human and animals cannot synthesize and therefore should only come from dietary intake, and thus, for the PCA analysis, we utilized the five EAA $\delta^{13}C$ values (Ile, Leu, Phe, Thr, Val) and five EAA $\delta^{15}N$ values (Ile, Leu, Phe, Thr, Val). Along with these five EAA $\delta^{13}C$ values, we also investigated the effectiveness of three source $\delta^{15}N$ values (Ser, Gly, Phe) in distinguishing each food group.

In conjunction with the PCA analysis, we used a Bayesian mixing model (MixSIAR) to estimate the contribution of sources to the whole diet based on five EAA $\delta^{13}C$ and $\delta^{15}N_{Phe}$ proxies. In the MixSIAR, EAA isotopic values of humans were input as raw data and sources were input as means and standard deviations. For the bulk isotope model, it is required to define the trophic discrimination factors (TDF) of each food source for the $\delta^{13}C$ and $\delta^{15}N$ values ($C_3$ plant: $\Delta^{13}C$ = 5.2‰ and $\Delta^{15}N$ = 3.8‰; $C_4$ plant: $\Delta^{13}C$ = 4.5‰ and $\Delta^{15}N$ = 3.8‰; animal: $\Delta^{13}C$ = 1‰ and $\Delta^{15}N$ = 3.8‰) [37]. However, in the EAA isotope model, we used a TDF of 0‰ for the EAA $\delta^{13}C$ and $\delta^{15}N_{Phe}$ values due to zero consumer-to-diet isotopic offsets. The EAA isotope model was run at length normal [41, 42]. The model is fit via Markov Chain Monte Carlo, which estimates entire posterior distributions for each variable, given the data. We set error structure as a multiplicative process. Data analyses were conducted using JAGS and R software [64]. All values in the text are given as mean ± standard deviation. The level of significance was set at $P \leq 0.05$.

## Results

### $\delta^{15}N_{AA}$ values

Nitrogen AA results for the human, animal, and cereal samples analyzed in this study are presented in S1-S3 Tables in S1 File. We obtained $\delta^{15}N_{AA}$ results for 10 bone collagen AAs from the human and animal samples: seven trophic AAs (Ala, Glx, Hyp, Pro, Val, Ile, Leu) and three source AAs (Phe, Gly, Ser) (S1-C and S2-C Tables in S1 File). For the cereals, 9 AAs were determined: six trophic AAs (Ala, Glx, Pro, Val, Ile, Leu) and three source AAs (Phe, Gly, Ser) (S3-C Table in S1 File). In order to assess quality control (QC), we used the relationship between proline and hydroxyproline since the hydroxylation of proline to form hydroxyproline does not involve exchange of nitrogen atoms [61]. Thus, a plot of $\delta^{15}N_{Hyp}$ values against $\delta^{15}N_{Pro}$ values is expected to have a slope close to 1 [65]. Our result revealed that the regression

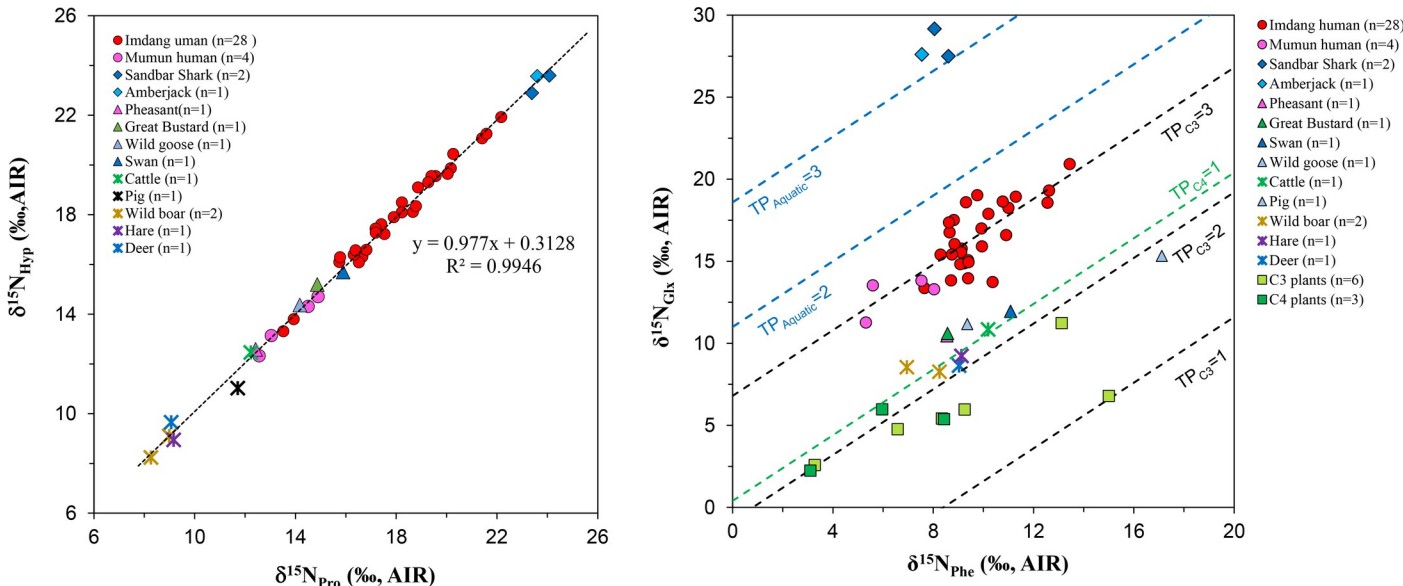

**Fig 3.** Quality control plot of $\delta^{15}N_{Pro}$ vs. $\delta^{15}N_{Hyp}$ values (A) for human and animal collagen and trophic position (TP) plot of compound specific nitrogen isotope values in phenylalanine and glutamic acid (B) of humans and animals from prehistoric Korean sites. The estimated TP lines are defined according to Chikaraishi et al. [17, 31]. Humans are located between TP of $C_3$ = 3 (black dot-line) and TP of Aquatic = 2 (blue dot-line) estimates suggesting that they might exploit both terrestrial and aquatic resources.

line of the $\delta^{15}N_{Hyp}$ vs. $\delta^{15}N_{Pro}$ values is close to $y = x$, demonstrating the excellent quality of the data ($R^2$ = 0.994; Fig 3A).

The $\delta^{15}N_{AA}$ values were applied to estimate trophic level (TP) of humans and associated animals, and assess which species were consumed by humans. Our results revealed the TP of humans and their food sources (Fig 3B). In TP estimates, we applied a β factor of -8.4‰ and $TDF_{Glx-Phe}$ value of 7.6‰ for the animals and terrestrial $C_3$ plants, but used a β factor of -0.4‰ and $TDF_{Glx-Phe}$ value of 7.6‰ for $C_4$ plants [17, 31]. The vertical shift between each line represents a different TP in Fig 3B. Estimated TP of modern cereals ($C_4$ plants) are lower than 1.0 but the TP of terrestrial $C_3$ plants is 1.5–2.0, and this is relatively higher than the TP of $C_3$ plants. Estimated TP of terrestrial herbivores (cattle, pig, wild boar, hare, deer) from the Mumun and Imdang sites are in agreement with their feeding habits. The terrestrial herbivores show a TP of 2.1–2.2, consistent with the expected values for pure herbivores (i.e. 2.0). Although livestock (cattle, pig) have $^{15}N$-enriched bulk collagen values, their TP (2.2) are similar to terrestrial herbivores, which suggests that the high bulk $\delta^{15}N$ results of the cattle and pigs is due to high $\delta^{15}N$ values of plants, likely grown by manuring. Game birds (pheasant, great bustard, wild goose, swan) also show similar TP values (1.9 to 2.4) consistent with terrestrial herbivorous diets, even though there is a 4.6‰ difference in the bulk $\delta^{15}N$ values among the game birds.

The humans from the Mumun and Imdang sites presented in this study have a wide range of TP values from 2.5 to 3.3. Bulk stable isotopic data showed that the Mumun humans have low bulk $\delta^{15}N$ values (6.8‰ to 7.6‰), and that the Imdang humans have a wide range of $\delta^{15}N$ values (7.6‰ to 13.3‰) [36, 37]. The Mumun humans had TPs that ranged between 2.5 to 2.9, and this indicates that the low bulk $\delta^{15}N$ values were likely caused by the low bulk $\delta^{15}N$ values of foods in the prehistoric environments. In contrast, the Imdang humans have a wider range of TP values (2.5 to 3.3), which could suggest the additional consumption of marine resources.

In general, terrestrial carnivores show $TP_{AA}$ values ranging from 2.9 to 3.1 [15]. Among the 28 Imdang individuals, 11 individuals have higher TP values (from 3.1 to 3.3) than the terrestrial carnivores (Fig 3B). Zooarchaeological studies reported that marine fish and shellfish were commonly identified at the site of Imdang, but not terrestrial carnivores [54–56]. Thus, we can assume that these 11 Imdang individuals having higher TP values than the carnivores could have consumed marine resources rather than terrestrial carnivores.

## $δ^{13}C_{AA}$ values

Carbon AA results of the humans, animals, and cereals are presented in S1-S3 Tables in S1 File. The $δ^{13}C_{AA}$ results of 13 bone collagen AAs, six EAAs (Ile, Leu, Lys, Phe, Thr, Val) and seven NEAAs (Ala, Asx, Glx, Gly, Hyp, Pro, Ser), from the human and animal samples are presented (S1-B and S2-B Tables in S1 File), and the $δ^{13}C_{AA}$ results of 13 plant amino acids, six EAAs (Ile, Leu, Phe, Thr, Val, Tyr) and seven NEAAs (Ala, Asx, Glx, Met, Gly, Pro, Ser), from the cereals are presented (S3-B Table in S1 File). For the quality control (QC) of the data, we tested the regression line of the $δ^{13}C_{Hyp}$ and $δ^{13}C_{Pro}$ values from the Mumun and Imdang sites, and it is close to a 1:1 line, indicating the excellent quality of the data ($R^2 = 0.969$) (Fig 4A).

Previous studies have shown that the $Δ^{13}C_{Gly-Phe}$ values are an effective proxy for discriminating between terrestrial and marine sources [6, 7, 11, 66]. The average $Δ^{13}C_{Gly-Phe}$ value for the Mumun humans was 12.4±1.2‰ and the average $Δ^{13}C_{Gly-Phe}$ value for the Imdang humans was 13.3±1.0‰, similar to terrestrial animals. Along with $Δ^{13}C_{Gly-Phe}$ values, it has been suggested that $Δ^{13}C_{Val-Phe}$ values are also effective at discriminating between terrestrial and aquatic sources [5, 11, 66]. Our $Δ^{13}C_{Val-Phe}$ results show that $Δ^{13}C_{Val-Phe}$ proxy can separate terrestrial from aquatic sources (Fig 4B). For the humans, the Mumun individuals have average $Δ^{13}C_{Val-Phe}$ values with a mean of -0.2‰ and the Imdang individuals have average $Δ^{13}C_{Val-Phe}$ values with a mean of 1.7‰, similar to terrestrial animals. In addition, $Δ^{13}C_{Val-Phe}$ can

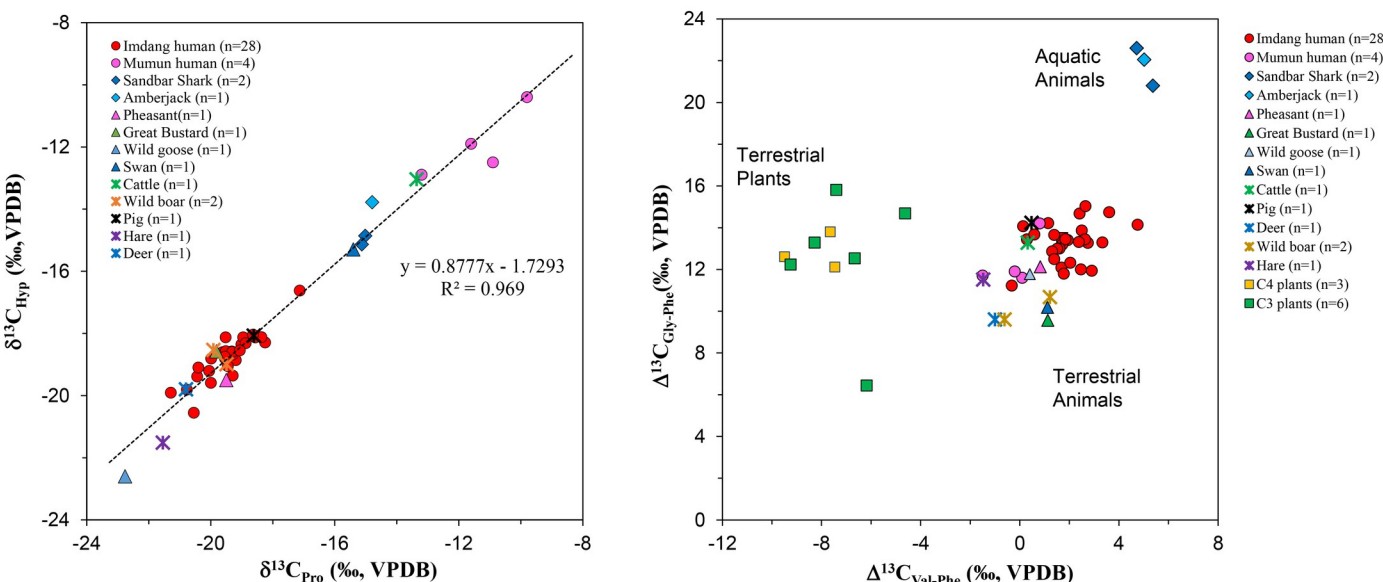

**Fig 4.** Quality control plot of $δ^{13}C_{Pro}$ vs. $δ^{13}C_{Hyp}$ values (A) for human and animal collagen, and $Δ^{13}C_{Gly-Phe}$ vs. $Δ^{13}C_{Val-Phe}$ of human and animal samples from Mumun and Imdang sites (B). All humans overlap with the terrestrial animals in $Δ^{13}C_{Gly-Phe}$ vs. $Δ^{13}C_{Val-Phe}$, suggesting consumption of terrestrial dietary sources.

separate terrestrial ($C_3$ and $C_4$) plants from terrestrial and aquatic animals. Thus, our $\Delta^{13}C_{Gly\text{-}Phe}$ and $\Delta^{13}C_{Val\text{-}Phe}$ proxies indicate that the most of dietary sources in the Mumun and Imdang humans are from terrestrial resources (Fig 4B), although the TP estimates reveals that several Imdang individuals (n = 11) consumed aquatic sources.

## Principal component analyses of EAAs

To assess the ability to detect differences in food groups, we conducted principal component analyses (PCA) using three options: 1. Five EAAs $\delta^{13}C$ values (Phe, Leu, Val, Ile, Thr) and four EAAs $\delta^{15}N$ values (Val, Leu, Ile, Phe); 2. Five EAA $\delta^{13}C$ values (Phe, Leu, Val, Ile, Thr) and three source AA $\delta^{15}N$ values (Ser, Gly, Phe); 3. Five EAA $\delta^{13}C$ values (Phe, Leu, Val, Ile, Thr) and one source AA $\delta^{15}N$ value (Phe) (S1 Fig in S1 File). Among the three PCA options tested here, PCA option 2 and 3, utilizing source AAs, exhibit a good separation and clustering of each food group (S1 Fig in S1 File). However, the most effective separation and clustering of each food source (marine, terrestrial, $C_3$, and $C_4$ plants) based on metabolic pathways was achieved through the combination of all five EAA $\delta^{13}C$ values (Phe, Leu, Val, Ile, Thr) and four $\delta^{15}N$ values (Val, Leu, Ile, Phe) (Fig 5). Detailed PCA data of nine EAAs from humans and food sources are shown in S4 Table in S1 File. The $\delta^{13}C$ values of five EAAs (Phe, Leu, Val, Ile, Thr) were significantly different between $C_3$ and $C_4$ plants. The $\delta^{15}N$ values of four EAAs (Val, Leu, Ile, Phe) were significantly different between marine and terrestrial animals. Both $\delta^{13}C$ and $\delta^{15}N$ values of EAAs can separate the terrestrial animals from other food sources. However, the PCA results showed that the EAA $\delta^{13}C$ and $\delta^{15}N$ values cannot discriminate between terrestrial herbivores and game birds due to their isotopic similarity in EAAs. To examine the EAA sources in each human, we plotted the Imdang and Mumun humans with each food group. We found that the Mumun humans clustered closely to the $C_4$ plants (Fig 5A), while the Imdang humans clustered closely to the game birds and terrestrial herbivores

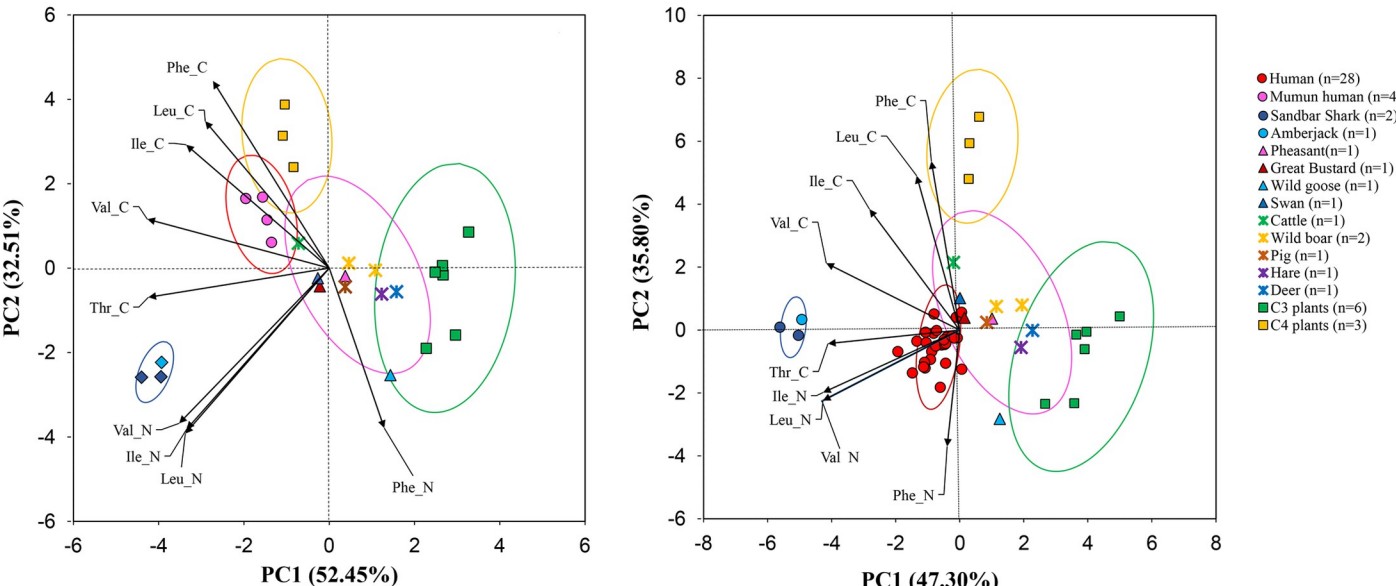

**Fig 5.** Score plots of principal component analyses (PCA) of $\delta^{13}C$ and $\delta^{15}N$ values of EAAs from humans and animals at the Mumun (A) and Imdang sites (B). The Mumun humans are clustered closely with $C_4$ plants and the Imdang humans are clustered closely with most of the terrestrial herbivores. Ellipses represent 95% standard deviation with arrows for EAAs being significant (p<0.05) correlation vectors. Game birds are not shown in these PCA figures due to overlap with terrestrial herbivores.

(Fig 5B). Thus, our PCA results show that the Imdang humans derived a large proportion of their EAAs from game birds and terrestrial herbivores, while the Mumun human derived a large proportion of their EAAs from $C_4$ plants.

## MixSIAR model

A MixSIAR model was conducted with five $\delta^{13}C_{AA}$ values (Leu, Val, Ile, Thr, Phe) and one source $\delta^{15}N_{AA}$ value (Phe). These six AAs were selected since they are only found in both the animals and cereals together, and they have little isotopic fractionation between diet and consumers [24, 25, 30]. Thus, as these six AAs are derived mainly from dietary sources, they can provide a more complete picture of the total diets of the Mumun and Imdang individuals. In S5 Table in S1 File, the trophic discrimination factors and the isotopic data of food sources for the MixSIAR model are listed. The EAA isotopic model output indicates that the Mumun humans had a high contribution from the $C_4$ plants (24–39%) and game birds (17–39%), and a small contribution from $C_3$ plants (3%) (S6 Table in S1 File). However, the Imdang humans had a high contribution from animal sources such as the game birds (17–63%), marine fish (8–46%), and a small contribution from $C_3$ (3–16%) and $C_4$ plants (3–8%). This indicates that the main EAA sources for the Mumun humans were $C_4$ plants and game birds (Fig 6A), whilst the main EAA sources for the Imdang humans were game birds, terrestrial animals, and marine fish (Fig 6B).

We examined if there is any difference in dietary estimates in the mixing models using the bulk and EAA isotope data. We generated and compared the MixSIAR model for both bulk and EAA isotopic values, respectively (S5 Table in S1 File). To compare the difference in dietary estimates between the two models, we used paired t-tests to evaluate possible differences in each food group. The t-test results showed that plant sources ($C_3$ and $C_4$ plants) are negatively correlated (P < 0.001) between the bulk and EAA isotope model, but animal sources (game birds and marine fish) are positively correlated (P < 0.001). This result indicates that the proportional contributions of plant sources ($C_3$ and $C_4$ plants) to both the Mumun and Imdang humans can be elevated in the bulk isotope model, and there is a tendency to overemphasize plant sources in the bulk isotope model (Fig 6). In contrast to plant sources, the

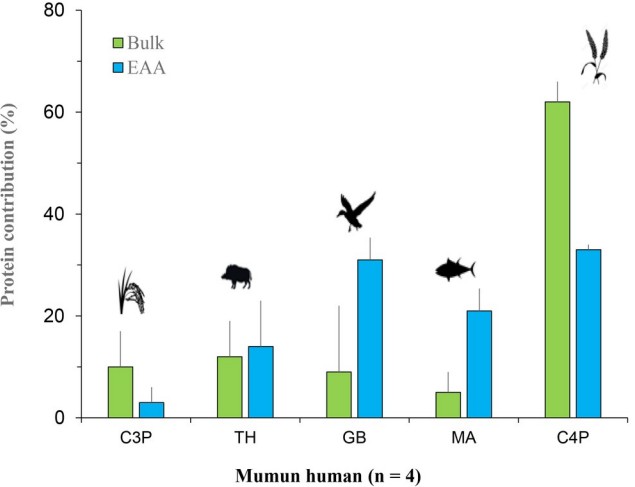
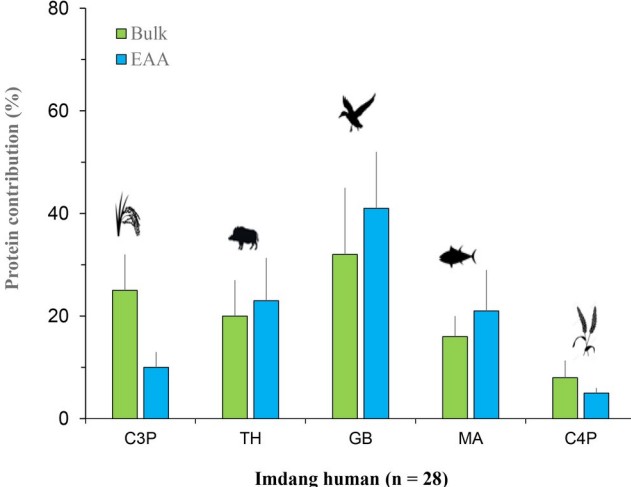

**Fig 6.** Model comparison between the dietary contribution of the five food groups to the Mumun (A) and Imdang humans (B) calculated by MixSIAR model. The dietary estimates on the EAA isotope model show that there is a tendency to reflect more contributions of animal sources (GB, MA), and less contributions of plants sources (C3P, C4P) than the bulk isotope model.

contribution of animal sources (game birds and marine fish) is increased when estimated by the EAA isotope model. This result indicates that the EAA isotope model could be more reflective of animal resources in dietary reconstruction than plant resources.

## Discussion

### Trophic position estimates

Our CSIA-AA study confirms that TP estimates can provide more accurate predictions of ancient human diets within the isotopic baselines of the surrounding environment. The TP values of the Mumun humans show that their low bulk $\delta^{15}$N values reflect the consumption of $\delta^{15}$N-low resources from the prehistoric ecosystem. On the other hand, the TP values of the Imdang humans show that the high bulk $\delta^{15}$N results of 11 of the Imdang individuals (TP 3.1 to 3.3) resulted from the consumption of $^{15}$N-enriched sources like marine fish. However, most of the Imdang individuals below TP 3.0 are associated with the consumption of terrestrial resources. In the trophic position estimates, TP values in humans were generated for theoretically 100% terrestrial and/or marine protein [16]. Practically, as omnivores, humans would be expected to consume multiple food sources consisting of marine fish, terrestrial animals, $C_3$ and $C_4$ plants. If humans had a mixed diet, we need to consider each β and TDF value from different food sources. Furthermore, there is a variation in β and TDF values in an ecological niche due to the different primary producers in ecosystems [4]. For example, the β value of $C_3$ plants is +8.4‰ and the β value of $C_4$ plants is -0.4‰ and the β value of marine plants -3.4‰ [32]. In addition, the commonly used $TDF_{Glx-Phe}$ of 7.6‰ is an average value of several observations in ecological studies and can vary between taxa [31]. Thus, it is required to establish more accurate β and TDF values of different food sources (e.g. β and TDF values of marine and $C_4$ plants). Furthermore, it is considered that the TP value of plants is around 1 [14, 15, 17] and the plant TP value is used as the basis for the TP estimation of humans and animals in the food webs. However, our results show that the TP estimates in grains (rice, soybean, adzuki bean, wheat, oat) are higher than 1, ranging from 1.0 to 2.1. Previously published data also show that the TP estimates in Danish wheat range from 1.4 to 1.7 [67] and German beans and pea range from 2.1 to 2.2 [68]. This discrepancy in plant TP values could be associated with the different lignin content between grains and leaves [69]. In dietary reconstruction, grains are the main crops consumed and could contribute more to human diets than any other plant tissues (i.e. leaves, stems, rachis). Unfortunately, most of the previous TP values of plants are mainly derived from the leaves. Thus, for more accurate TP estimates on human diets, it is necessary to have more detailed studies to calculate the TP values of grains among plant species [69, 70].

### Amino acid proxies and PCA patterns

For $\delta^{13}$C proxies, previous studies suggested that two AA proxies ($\Delta^{13}C_{Gly-Phe}$ and $\Delta^{13}C_{Val-Phe}$) are effective at discriminating between terrestrial and aquatic sources [7, 11, 66]. However, our study shows that the $\Delta^{13}C_{Gly-Phe}$ and $\Delta^{13}C_{Val-Phe}$ results are incompatible with the estimated trophic position (TP) using $\Delta^{15}N$ $AA_{Glx-Phe}$ values. This inconsistency indicates that the two $\delta^{13}C_{AA}$ proxies are less powerful to fully separate between terrestrial and marine sources when humans have mixed diets. This means that one or two amino acid markers (e.g. Gly, Val, Phe) are less likely to distinguish dietary sources in mixed human diets. Instead, our PCA data with EAAs provide an improved clustering and separation of $C_3$, $C_4$, marine and terrestrial protein consumers (Fig 5), and better information on the main source of dietary proteins. The $\delta^{13}$C and $\delta^{15}$N values of EAAs in PCA, instead of source $\delta^{15}N_{AA}$ values, can well separate all four food sources according to their metabolic pathways. Our PCA patterns show that the Mumun

humans are close to the $C_4$ plants, suggesting that $C_4$ plants played a significant role in Mumun diets. In contrast, the Imdang humans cluster with the game birds and terrestrial herbivores, indicating that the hunting of wild animals was important in their subsistence activities. Additionally, we found that the EAA patterns in food sources are not able to discriminate the confounding isotope values between terrestrial herbivores and game birds at the Imdang site. This is associated with their isotopic similarity between these two animal groups and their similarity in dietary sources ($C_3$-based diets) in the terrestrial food webs.

## Comparison of mixing models

The output of the two MixSIAR models showed that the model predictions based on both bulk and EAA isotope values yielded different palaeodietary patterns. In the Mumun, the bulk isotope model showed high contributions of $C_4$ plants and a low contribution of other sources, emphasizing the consumption of $C_4$ plants (~62%). In contrast, the EAA isotope model finds a contribution of other animal proteins in $C_4$ plant-dominated Mumun diets (Fig 6). For the Imdang, the bulk isotope model showed the high contribution of game birds (32%) and low contribution of $C_4$ plants (8%). In particular, the EAA isotope model showed a much higher contribution of animal proteins than the bulk isotope model (Fig 6), emphasizing more the consumption of animal sources such as game birds (41%) and marine animals (21%). Thus, our study confirms that the EAA isotope model can reflect more the consumption of animal proteins in mixed human diets.

This discrepancy between the bulk and EAA isotope models is likely due to variability in the extent of routing of EAAs in foodstuffs. Generally, plants have lower amounts of protein and higher contents of carbohydrates than animal resources. Although there was the same contribution from both plant and animal proteins in human diets, EAAs in human bone collagen could be routed firstly from the EAA in animal proteins rather than plant proteins [71]. This is due to the fact that animal proteins require less metabolic expense in EAA routing than plants proteins. Therefore, the EAA isotope model is more related to the consumption of animal proteins in human diets. On the other hand, the bulk isotope model emphasizes the relative importance of one or two sources, highlighting the dependence on plant proteins. A recent study using the EAA isotope model also suggested that the contribution of animal proteins (marine fish) to total dietary proteins in ancient Romans can be underrepresented in bulk isotope data [14]. Our EAA isotope model in this study shows that the contribution of plant proteins to total dietary proteins can be overrepresented in bulk isotope data. Thus, when humans have mixed diets, the EAA isotope model can be much more suited to distinguishing the consumption of animal proteins in human diets than the bulk isotope model.

## Diet of the prehistoric Korean populations

The CSIA-AA approach finds a difference in relative contributions of each dietary source for the two prehistoric Korean populations. In terms of plant sources, the Mumun had high contributions of EAAs from $C_4$ plants (Fig 7A), but Imdang people had high contributions of EAAs from $C_3$ plants to their total diets. This indicates that these two prehistoric Korean populations might have cultivated different crops: $C_4$ plants for the Mumun and $C_3$ plants for the Imdang, even though the subsistence economy in both populations was based on agriculture. Previous work has suggested that early agriculture on the Korean peninsula was established during the Mumun period (1500–100 BC), but there is no clear evidence which grains (rice *vs.* millet) were first cultivated as the main crops during that period [38]. However, our study shows that $C_4$ plants (millets) were the main staples during the Mumun period, supporting the possibility that millets could be the first crops in prehistoric Korea. At the Imdang sites dated

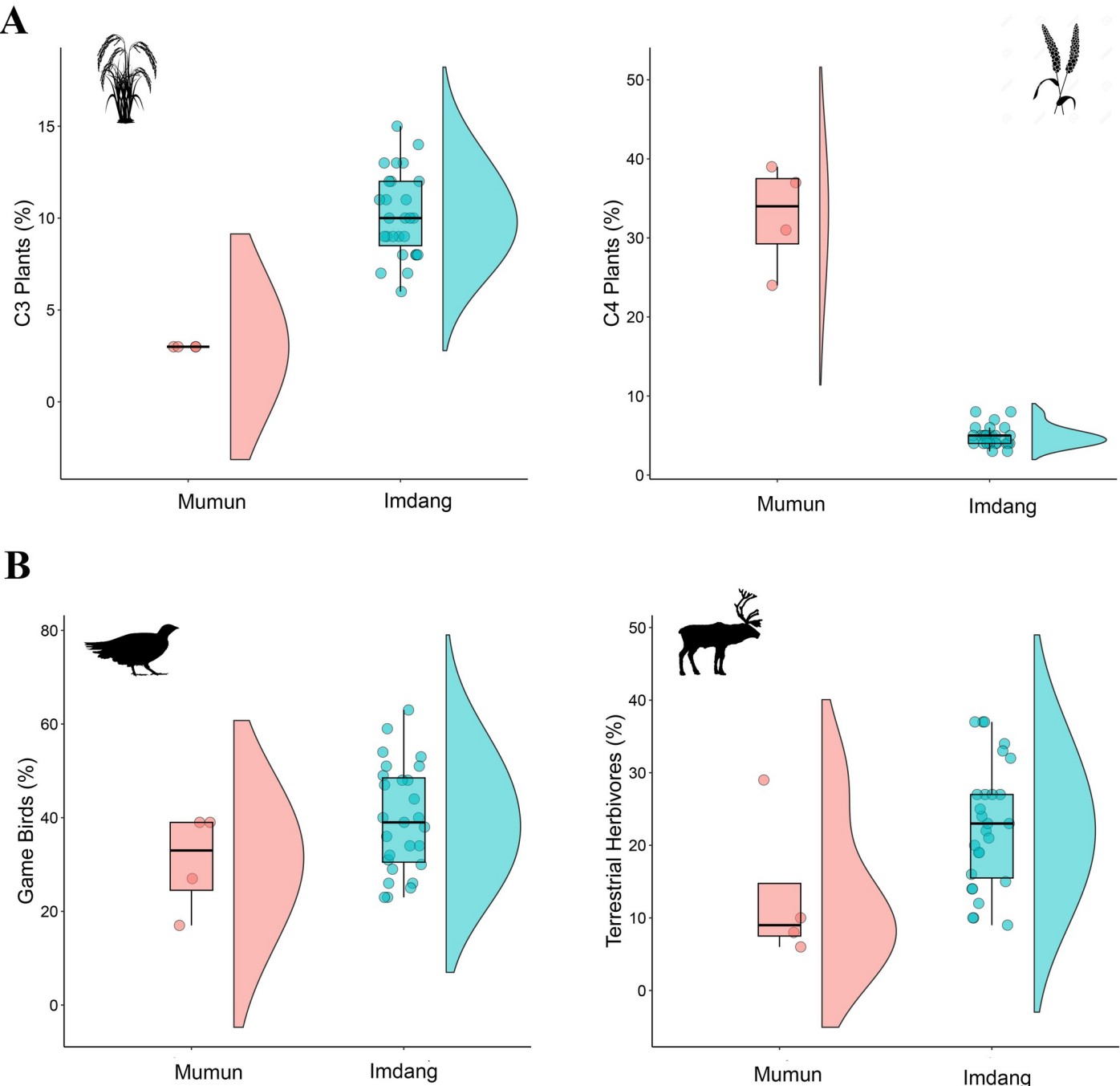

**Fig 7.** Raincloud plots of dietary estimates of plant (A) and animal (B) proteins generated by the EAA isotope model. There is a clear difference in dietary contributions of plant (C$_3$ vs. C$_4$) and animal (game birds vs. terrestrial herbivores) sources between the two prehistoric Korean populations.

to Proto-Three Kingdom period, C$_3$ plants (rice, barley, soybean) were the dominant plant resources for the human diets (Fig 7A). In Korean archaeology, it has been suggested that rice agriculture was introduced earlier and that it intensified later during the Proto-Three Kingdom period (BC 108–313 AD) [44, 47, 52]. Our EAA isotope data suggest that agriculture was based on C$_3$ plants (likely rice) during the Proto-Three Kingdom period. Therefore, despite

the problem of sample size in this study, it is possible to speculate that there was a change in the main crops cultivated from the Bronze Age to the Proto-Three Kingdom period in Korea.

In terms of animals consumed, the Imdang people had high contributions from animal sources to their total diets, while the Mumun people had relatively low contributions from animal sources (Fig 7B). This difference indicates that animal proteins were not mainly consumed until the Mumun period, but in the later Proto-Three Kingdom period, they became significant food items in human diets. It has previously been suggested that most of the domestic animals (i.e. cattle, pig, horse, chicken) became important food sources in human diets during the Proto-Three Kingdom period (BC 108–313 AD) [39]. However, our study reveals that most of the animal resources in the Imdang diets were coming from the hunting of wild birds and herbivores, as well as from fishing (Fig 7B). In relative terms, wild terrestrial birds (e.g. pheasant, wild goose) contributed the most animal protein sources to human diets. Therefore, we can postulate that hunting wild animals and fishing still played a significant role in subsistence activities during the Proto-Three Kingdom period, even though the subsistence economy was based on plant cultivation and animal husbandry.

## Conclusions

We applied CSIA-AA to bone collagen to enhance palaeodietary reconstructions in South Korean prehistoric populations. We measured $\delta^{13}C_{AA}$ and $\delta^{15}N_{AA}$ values in human and animal bone collagen from the archaeological sites, and modern cereals from South Korea as reference materials. CSIA-AA provided new information about Prehistoric Korean subsistence strategies, the spread of early agriculture based on the domesticated grains and the subsistence economy of ancient states from the two different-era sites. Results of estimated trophic positions (TP) show that the two prehistoric populations (Mumun and Imdang) obtained most of their dietary protein from terrestrial sources. However, the EAA proxies and PCA data revealed that there were different dietary patterns between the Mumun to the Proto-Three Kingdom periods. The humans from the Mumun sites consumed primarily $C_4$ plant resources, while the Imdang humans consumed primarily terrestrial animals. Quantitative estimates through the MixSIAR also found a variation in relative contributions of each food source to human diets in these two prehistoric populations. The EAA isotope model revealed that the Mumun had a high dietary contribution from $C_4$ plants. However, the Imdang had a high dietary contribution of the $C_3$ plants and a wide range of contributions from animal sources such as game birds, terrestrial mammals and marine animals. In this study, our CSIA-AA approach shows that the two investigated prehistoric Korean cultures differed in terms of the main crops consumed and that hunting and fishing in prehistoric Korea were still important subsistence activities along with agriculture in the Proto-Three Kingdom period. Finally, this study demonstrates that the use of CSIA-AA and MixSIAR to reconstruct palaeodiets allows us to explore more accurately the contribution of each dietary source to whole human diets and enhance our understanding of human adaptability to environments in prehistoric Korea.

## Supporting information

**S1 File.**
(PDF)

**S1 Data.**
(ZIP)

## Acknowledgments

We are grateful to In-Seong Cheong, the director at the Yeungnam University Museum and Soyoung Lee, curator at the Chungbuk National University Museum for access to skeletal materials. We are also grateful to Prof. Kyung-Hoon Shin and Dr. Hyuntae Choi for analytical help at the Department of Marine Sciences, Hanyang University. We would like to thank Zuzana Hansen, and Anastasia Brozou for help with sample preparation and collagen extraction at the Moesgaard Archaeo-Science Laboratory (Aarhus University).

## Author Contributions

**Conceptualization:** Kyungcheol Choy, Hee Young Yun.

**Formal analysis:** Kyungcheol Choy, Hee Young Yun.

**Funding acquisition:** Kyungcheol Choy.

**Investigation:** Kyungcheol Choy, Hee Young Yun.

**Methodology:** Kyungcheol Choy, Hee Young Yun, Benjamin T. Fuller, Marcello A. Mannino.

**Project administration:** Kyungcheol Choy, Hee Young Yun.

**Visualization:** Kyungcheol Choy, Hee Young Yun.

**Writing – original draft:** Kyungcheol Choy, Hee Young Yun.

**Writing – review & editing:** Kyungcheol Choy, Hee Young Yun, Benjamin T. Fuller, Marcello A. Mannino.

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
