## [Decision Letter · Decision Letter 0]

8 Jan 2024

PONE-D-23-38093Enhanced dietary reconstruction of Korean prehistoric populations by combining d 13 C and d 15 N amino acids of bone collagenPLOS ONE

Dear Dr. Choy,

Thank you for submitting your manuscript to PLOS ONE. After careful consideration, we feel that it has merit but does not fully meet PLOS ONE’s publication criteria as it currently stands. Therefore, we invite you to submit a revised version of the manuscript that addresses the points raised during the review process.

We look forward to receiving your revised manuscript.

Kind regards,

John P. Hart, Ph.D.

Academic Editor

PLOS ONE

Journal Requirements:

2. In your manuscript, please provide additional information regarding the specimens used in your study. Ensure that you have reported human remain specimen numbers and complete repository information, including museum name and geographic location. 

For more information on PLOS ONE's requirements for paleontology and archeology research, see https://journals.plos.org/plosone/s/submission-guidelines#loc-paleontology-and-archaeology-research.

"This work was supported by Research Funds for Young Faculty Club at Hanyang University (HY-2018-00000001315) and also supported by the National Research Foundation of Korea (NRF) funded by the Ministry of Education (NRF-2022S1A5A2A03051382)."

Additional Editor Comments:

Both reviewers are positive about your work. However, both indicate the need for revision to clarify various sampling, methodological, statistical, and interpretive issues. Please address all of the comments and suggestions while making your revisions.

Reviewers' comments:

Reviewer's Responses to Questions

**Comments to the Author**

1. Is the manuscript technically sound, and do the data support the conclusions?

Reviewer #1: Yes

Reviewer #2: Yes

2. Has the statistical analysis been performed appropriately and rigorously? 

Reviewer #1: Yes

Reviewer #2: Yes

3. Have the authors made all data underlying the findings in their manuscript fully available?

Reviewer #1: No

Reviewer #2: Yes

4. Is the manuscript presented in an intelligible fashion and written in standard English?

Reviewer #1: Yes

Reviewer #2: Yes

5. Review Comments to the Author

Reviewer #1: Reviewer comments to the authors

General comments

Dear authors, your article presents highly valuable data for the dietary reconstruction of Korean prehistoric populations. The use of Compound-Specific Stable Isotope Analysis from Amino Acids (CSIA-AA) distinctly strengthens this study and contributes significant insights to the published pool of CSIA-AA data in archaeology. This, in turn, optimises and enriches prospects for future research in the field of dietary reconstructions.

I extend my congratulations to the authors for conducting such an exemplary study. Additionally, I would like to suggest minor revisions that could potentially further enhance the work for its publication.

Furthermore, I encourage the authors to report the scripts for the PCA and the MixSIAR models. Sharing this information would significantly contribute to advancing the use of CSIA-AA in this field and optimise its interpretation.

Minor revisions

- Page 3 (lines 64-66), to the comment “Amino acids (AAs) can be divided into two groups on the basis of their synthesis and function: essential amino acid (EAA) and non-essential amino acid (NEAA)”:

I would like to suggest to make clear that this distinction is related to the carbon pathways and that it's different for nitrogen, for which they should rather differentiate between source and trophic amino acids (see e.g. O'Connell (2017) on the nitrogen metabolic pool).

O'Connell TC. (2017) 'Trophic' and 'source' amino acids in trophic estimation: a likely metabolic explanation. Oecologia. 184(2):317-326. doi: 10.1007/s00442-017-3881-9.

- Page 8 (line 216), about derivatisation method(s?):

It's important to note that the derivatisation process applies to the actual amino acid, not the isotope value itself. However, I'm unclear on whether you employed two distinct derivatisation approaches—one for measuring nitrogen isotope values and another for carbon. Your description suggests this might be the case, but it's not entirely clear, especially regarding the method used for carbon isotope measurements. Could you please provide clarification on this aspect?

- Page 10 (lines 273 - 274), to the sentence: “Standard deviations of carbon isotope measurements were generally ±0.4‰ as determined by repeated injections of the 10 AA standard mixture”:

Have you considered propagating the error due to inclusion of exogenous carbon from your solvents? I believe that's something important to do. See Docherty et al (2001)

Docherty G, Jones V, Evershed RP. Practical and theoretical considerations in the gas chromatography/combustion/isotope ratio mass spectrometry delta(13)C analysis of small polyfunctional compounds. Rapid Commun Mass Spectrom. 2001;15(9):730-8. doi: 10.1002/rcm.270. PMID: 11319796.

- Page 11 (line 312), regarding: “four essential (Ile, Leu, Phe, Val)”:

Please, be cautious regarding leucine, isoleucine, and valine, as they do not exhibit typical behaviour as 'essential' for nitrogen. Previous observations have noted fractionation in these amino acids (indicating their role as trophic amino acids). In all three cases, the diet-to-consumer offset was observed to be quite high, approximately 5 permille, with a significant level of uncertainty. For more insights, refer to experimental studies such as McMahon and McCarthy (2016).

McMahon, K.W. and McCarthy, M.D., 2016. Embracing variability in amino acid δ15N fractionation: mechanisms, implications, and applications for trophic ecology. Ecosphere, 7(12), p.e01511.

- Page 11 (lines 315-317), regarding: “In order to assess quality control (QC), we used the relationship between proline and hydroxyproline since the hydroxylation of proline to form hydroxyproline does not involve exchange of nitrogen atoms [56]”.

You should be careful here, as you only consider the quality of proline and hydroxyproline. Ideally, you should estimate the d13C and d15N bulk stable isotope values from the amino acid values and compare them with your measured bulk SIA values. See e.g. Fontanals-Coll, et al. (2023), but also older research by Styring and Howland. I consider this really important to assess the quality of your research, please try to address.

Fontanals-Coll M, Soncin S, Talbot HM, von Tersch M, Gibaja JF, Colonese AC, Craig OE. Stable isotope analyses of amino acids reveal the importance of aquatic resources to Mediterranean coastal hunter-gatherers. Proc Biol Sci. 2023 Feb 22;290(1993):20221330. doi: 10.1098/rspb.2022.1330.

- Page 13 (lines 379 - 380), regarding “Previous studies have shown that the d13CGly-Phe values are an effective proxy for discriminating between terrestrial and marine sources [6, 7, 11, 58]”.

The figure does not provide an explanation of how the shaded regions, within which terrestrial and aquatic values are expected to fall, were created. I assume you utilised data from these studies? If so, please make this explicit, perhaps in the figure's caption, to enhance clarity.

- Page 13 (lines 385-386), about: “Our d13CVal-Phe results show that d13CVal-Phe can separate terrestrial from aquatic sources.”

Figure? or Figure 4B could be: axis x: Gly-Phe, axis y: Val-Phe rather than bulk d15N as it has been previously done.

- Page 14 (lines 394-396), regarding: “To assess differences in food groups, we carried out principal component analyses (PCA) with five d13C (Phe, Leu, Val, Ile, Thr) and four d15N (Val, Leu, Ile, Phe) values of essential amino acids.”

Consistency is crucial here. Although you decided to run the PCA with essential amino acids, you included trophic amino acids. You should exclude Val, Leu, and Ile for their stable nitrogen isotope values for this. I suggest to re-run the PCA and decide between two different options: a) Repeat the analysis but excluding d15N for Ile, Leu, and Val, or b) Consider the inclusion of all (essential, non-essential, trophic) amino acids. Both approaches will provide different insights; however, the first option (a), while not altering the results significantly, ensures consistency for readers and researchers in the field.

- Page 15 (lines 419-420), regarding: “A MixSIAR model was conducted with five d13CAA values (Leu, Val, Ile, Thr, Phe) and one d15NAA value (Phe).”

I would like to once again encourage the authors to publish the scripts. It's not entirely clear how you constructed your MixSIAR model. Did you consistently take into account the calorific or protein contribution of different foodstuffs? For instance, did you consider the protein contribution of cereals, assuming 70% carbohydrates and 30% proteins? If so, I am satisfied with the model. However, I believe that, like me, other researchers would appreciate having this information available. This isn't just for the purpose of assessing your work but also for the benefit of learning and improving their own methodologies and interpretations , especially considering that CSIA-AA is still cutting-edge.

Reviewer #2: This research paper presents bulk collagen and amino acid stable isotope data (carbon and nitrogen) from bone material (human and animal) of four sites in South Korea. The four sites are Prehistoric cemeteries and represents different chronocultural periods. Four human bones and 4 animal bones are spread on three sites of the Mumum period (Bonze Age culture for this geographical area) and 28 human bones along with 11 animal remains from different species (diversified and representing the resources potentially consumed) are from a site with 1600 burials of the Proto-Three kingdom period (Imdang, more recent than Mumum). The archaeological material is enhanced with modern local domesticated plants (n=9, C3 and C4 cereals) to create a local isotopic baseline. Bulk and CSIA-AA data are discussed thanks to statistical analyses (PCA) and Bayesian models. Results indicate (1) different dietary interpretations when comparing bulk and AA data and (2) show how isotopic data improve the knowledges about Prehistoric Korea subsistence strategies, still underdeveloped in the scientific community. The paper deserves publication for both aspects of methodological considerations and archaeological information provided.

Specific comments are proposed below

P 3, l 72 and 73: provide the trophic step for both Glu and Phe

P4, l102-104: late in the manuscript one can understand the real interest to use isotopic analysis to study diet, but this sentence deserves more information about the archaeological issue in this specific context

P4, l113: did you correct the C isotopic data (Suess effect)? if yes mention it in the paper, if not explain why.

P5 l114-116: better explain the archaeological issue of this study

P5: Mumum material: explain potential difficulty to get material and the exact anatomical part sampled. Why only 4 remains on 3 sites? how it can reflect a pattern or if not explain better how it improves the previous knowledge (to add also in the conclusion part).

The same for Imdang. Explain it is coming from scarified humans and how they can reflect a pattern and if not how it improves the previous knowledge (to add also in the conclusion part).

L134-136: push forward this issue and also highlight it in the conclusion.

P8-9: all the technical part should go in a Supplementary Material file

P11, l306: P>0.05?

P14, l405-409: it seems that it is twice the same sentence; check and rewrite what is necessary

P20 conclusion: add sentences about the initial archaeological issues (cf previous comments) and more explain the archaeological issues about agriculture and game in these two different periods. Explain why these two periods are interesting to compare and explain how despite the very small sample size and the specific origin of the material, the significance of the isotopic data (both bulk collagen and AA).

6. PLOS authors have the option to publish the peer review history of their article (what does this mean?). If published, this will include your full peer review and any attached files.

Reviewer #1: **Yes: **Maria Fontanals-Coll

Reviewer #2: No

---

## [Author Response · Author response to Decision Letter 0]

15 Feb 2024

5. Reviewer Comments: #Reviewer 1

General comments

Dear authors, your article presents highly valuable data for the dietary reconstruction of Korean prehistoric populations. The use of Compound-Specific Stable Isotope Analysis from Amino Acids (CSIA-AA) distinctly strengthens this study and contributes significant insights to the published pool of CSIA-AA data in archaeology. This, in turn, optimizes and enriches prospects for future research in the field of dietary reconstructions. I extend my congratulations to the authors for conducting such an exemplary study. Additionally, I would like to suggest minor revisions that could potentially further enhance the work for its publication. Furthermore, I encourage the authors to report the scripts for the PCA and the MixSIAR models. Sharing this information would significantly contribute to advancing the use of CSIA-AA in this field and optimize its interpretation.

Minor revisions.

We thank the reviewer for their comments. We agree with the reviewer and have uploaded the R scripts for the PCA and the MixSIAR models in Supplementary Information. We added the R code for all MixSIAR (pdf file) and imported outcomes in supplementary information. This includes data files of humans, five food groups, discrimination factor, and final MixSIAR results. We uploaded these raw data as excel files.

Minor revisions

- Page 3 (lines 64-66), to the comment “Amino acids (AAs) can be divided into two groups on the basis of their synthesis and function: essential amino acid (EAA) and non-essential amino acid (NEAA)”: I would like to suggest to make clear that this distinction is related to the carbon pathways and that it's different for nitrogen, for which they should rather differentiate between source and trophic amino acids (see e.g. O'Connell (2017) on the nitrogen metabolic pool).

We added these sentences: 

“While this constitutes the conventional definition of essential (EAA) and nonessential amino acids (NEAA), this definition is not solely confined to dietary considerations but is also extended to include the role of amino acids in supporting protein deposition and growth during different stages of life [20]. Only the carbon isotope values in AA metabolism are neatly aligned with the conventional distinction between EAA and NEAA.”

We also added these sentences: 

In contrast to �13C values in AA pathways, �15N AA values do not conform to the conventional definition of EAA and NEAA. Instead, there has been the development of the novel definition of source (Tyr, Lys, Phe, Ser, Gly), metabolic (Thr) and trophic AAs (Pro, Ala, Glx, Val, Leu, Asx) in nitrogen AAs [24-28]. However, it is important to be cautious when applying the meaning of source and trophic AAs in nitrogen to dietary studies as these definitions can be variable due to the species studied and developmental age. For instance, among the known source AAs (Tyr, Lys, Phe, Ser, Gly) in nitrogen, glycine and serine have been found to have a diet-to-consumer offset in pigs (�15N Ser: 3.2‰ in C3 diets and 2.6‰ in C4 diets; �15NGly: 0.8‰ in C3 diets and 2.2‰ in C4 diets) [29]. Only phenylalanine (Phe) appears to exhibit a relative ~0‰ diet-to-consumer offset in many species including humans [26, 30, 31]. Thus, we cannot directly substitute source AAs to EAAs for all animal species and humans, and more research on the definition of source, trophic and metabolic AAs is necessary.

We also added these references:

Lübcker N, Whiteman JP, Shipley ON, Hobson KA, Newsome SD. Use of amino acid isotope analysis to investigate capital versus income breeding strategies in migratory avian species. Methods Ecol. Evol. 2023;00: 1-14

McMahon KW, McCarthy MD. Embracing variability in amino acid d15N fractionation: mechanisms, implications, and applications for trophic ecology. Ecosphere. 2016;7: e01511

O'Connell TC. 'Trophic' and 'source' amino acids in trophic estimation: a likely metabolic explanation. Oecologia. 2017;184: 317-326.

Yun HY, Larsen T, Choi B, Won E, Shin K. Amino acid nitrogen and carbon isotope data: potential and implications for ecological studies. Ecol. Evol. 2022;12: e8929

Fuller BT, Petzke KJ. The dietary protein paradox and threonine 15N-depletion: pyridoxal-5’- phosphate enzyme activity as a mechanism for the d15N trophic level effect. Rapid Commun. Mass Spectrom. 2017; 31:705-718

Hare PE, Fogel ML, Stafford Jr TW, Mitchell AD, Hoering TC. The isotopic composition of carbon and nitrogen in individual amino acids isolated from modern and fossil proteins. J Archaeol Sci. 1991; 18: 277-292.

- Page 8 (line 216), about derivatisation method(s?): It's important to note that the derivatisation process applies to the actual amino acid, not the isotope value itself. However, I'm unclear on whether you employed two distinct derivatisation approaches—one for measuring nitrogen isotope values and another for carbon. Your description suggests this might be the case, but it's not entirely clear, especially regarding the method used for carbon isotope measurements. Could you please provide clarification on this aspect?

This has been clarified in the text. 

- Page 10 (lines 273 - 274), to the sentence: “Standard deviations of carbon isotope measurements were generally ±0.4‰ as determined by repeated injections of the 10 AA standard mixture”: Have you considered propagating the error due to inclusion of exogenous carbon from your solvents? I believe that's something important to do. See Docherty et al (2001).

Yes, this was considered and the sentence was clarified: “Standard deviations of carbon isotope measurements were generally ±0.4‰ as determined by repeated injections of the 10 AA standard mixture and each measured mean amino acid �13C was corrected relative to the mean amino acid �13C of standards to account for the exogenous carbon and kinetic fractionation introduced during derivatization [60].”

- Page 11 (line 312), regarding: “four essential (Ile, Leu, Phe, Val)”: Please, be cautious regarding leucine, isoleucine, and valine, as they do not exhibit typical behavior as 'essential' for nitrogen. Previous observations have noted fractionation in these amino acids (indicating their role as trophic amino acids). In all three cases, the diet-to-consumer offset was observed to be quite high, approximately 5 permil, with a significant level of uncertainty. For more insights, refer to experimental studies such as McMahon and McCarthy (2016). McMahon, K.W. and McCarthy, M.D., 2016. Embracing variability in amino acid δ15N fractionation: mechanisms, implications, and applications for trophic ecology. Ecosphere, 7(12), p.e01511.

This was changed to source and trophic amino acids. 

In this study, we measured 10 amino acids for nitrogen (Ala, Gly, Val, Leu, Ile, Pro, Ser, Glx, Phe, Hyp). Three essential (Ile, Leu, Val) amino acids have quite high diet to consumer offsets. Only one source amino acid in nitrogen (Phe) has a zero offset. Thus, for PCA analysis, the use of only Phe in nitrogen does not have enough power to separate the 5 food sources and we tried to look at the EAA separation between species and a strong separation in food sources was found.

Thus, we followed the conventional definition of EAAs. For MixSIAR modeling later, we used one source amino acid in nitrogen (Phe) to estimate for dietary contribution by MixSIAR modeling due to zero offset. 

- Page 11 (lines 315-317), regarding: “In order to assess quality control (QC), we used the relationship between proline and hydroxyproline since the hydroxylation of proline to form hydroxyproline does not involve exchange of nitrogen atoms [56]”. You should be careful here, as you only consider the quality of proline and hydroxyproline. Ideally, you should estimate the d13C and d15N bulk stable isotope values from the amino acid values and compare them with your measured bulk SIA values. See e.g. Fontanals-Coll, et al. (2023), but also older research by Styring and Howland. I consider this really important to assess the quality of your research, please try to address. Fontanals-Coll M, Soncin S, Talbot HM, von Tersch M, Gibaja JF, Colonese AC, Craig OE. Stable isotope analyses of amino acids reveal the importance of aquatic resources to Mediterranean coastal hunter-gatherers. Proc Biol Sci. 2023 Feb 22;290(1993):20221330. doi: 10.1098/rspb.2022.1330.

This was done. We made multiple measurement of Standard AAs and add this sentence: “Analytical error (SE) of �13CAA measurements was estimated from the AA standards, and ranged from 0.10‰ for �13C Val to 0.45‰ for �13C Phe”. 

- Page 13 (lines 379 - 380), regarding “Previous studies have shown that the d13CGly-Phe values are an effective proxy for discriminating between terrestrial and marine sources [6, 7, 11, 58]”. The figure does not provide an explanation of how the shaded regions, within which terrestrial and aquatic values are expected to fall, were created. I assume you utilized data from these studies? If so, please make this explicit, perhaps in the figure's caption, to enhance clarity.

We referred to the previous studies. Previous studies showed that �13CGly-Phe can separate clearly aquatic from terrestrial animals. However, if humans and animals have a mixed diet, �13CGly-Phe cannot show a clear separation between aquatic and terrestrial animals. That’s why there are shaded regions in Figure 4B.

- Page 13 (lines 385-386), about: “Our d13CVal-Phe results show that d13CVal-Phe can separate terrestrial from aquatic sources.” Figure? or Figure 4B could be: axis x: Gly-Phe, axis y: Val-Phe rather than bulk d15N as it has been previously done.

We changed Figure 4B with �13Cval vs��13Cphe .

- Page 14 (lines 394-396), regarding: “To assess differences in food groups, we carried out principal component analyses (PCA) with five d13C (Phe, Leu, Val, Ile, Thr) and four d15N (Val, Leu, Ile, Phe) values of essential amino acids.” Consistency is crucial here. Although you decided to run the PCA with essential amino acids, you included trophic amino acids. You should exclude Val, Leu, and Ile for their stable nitrogen isotope values for this. I suggest to re-run the PCA and decide between two different options: a) Repeat the analysis but excluding d15N for Ile, Leu, and Val, or b) Consider the inclusion of all (essential, non-essential, trophic) amino acids. Both approaches will provide different insights; however, the first option (a), while not altering the results significantly, ensures consistency for readers and researchers in the field.

We tested all three options:

1. Only EAA (five �13C (Phe, Leu, Val, Ile, Thr) and four �15N (Val, Leu, Ile, Phe),

2. Five EAA (Phe, Leu, Val, Ile, Thr� and three source (Ser, Gly, Phe),

3. Five d13C (Phe, Leu, Val, Ile, Thr) and one Source AA (Phe),

We found that: 1. Only EAA can be best separated depending on food group: marine, terrestrial, C3 plants, and C4 plants. This means that only EAAs (combining five �13C variables of Phe, Leu, Val, Ile, Thr and four �15N (Val, Leu, Ile, Phe EAAs) have a strong power to differentiate food group (plant and animal food sources) and to attain the best separation of each food group. The purpose of PCA is a separation of each food source and to show the relationship between food sources and human diets. The other two options separate each food group in a less satisfactory fashion.

Thus, we added the following for clarification:

To assess the ability to detect differences in food groups, we conducted principal component analyses (PCA) using three options: 1. Five EAAs �13C values (Phe, Leu, Val, Ile, Thr) and four EAAs �15N values (Val, Leu, Ile, Phe); 2. Five EAA �13C values (Phe, Leu, Val, Ile, Thr) and three source AA �15N values (Ser, Gly, Phe); 3. Five EAA �13C values (Phe, Leu, Val, Ile, Thr) and one source AA �15N value (Phe) (Fig. S1). Among the three PCA options tested here, PCA option 2 and 3, utilizing source AAs, exhibit a good separation and clustering of each food group (Fig. S1). However, the most effective separation and clustering of each food source (marine, terrestrial, C3, and C4 plants) based on metabolic pathways was achieved through the combination of all five EAA δ13C values (Phe, Leu, Val, Ile, Thr) and four δ15N values (Val, Leu, Ile, Phe)

 Additionally, we added the detailed Figure S1 in Supplementary information (Fig S1).

- Page 15 (lines 419-420), regarding: “A MixSIAR model was conducted with five d13CAA values (Leu, Val, Ile, Thr, Phe) and one d15NAA value (Phe).”

I would like to once again encourage the authors to publish the scripts. It's not entirely clear how you constructed your MixSIAR model. 

We added the R codes in the Supplementary Information.

Did you consistently take into account the calorific or protein contribution of different foodstuffs? For instance, did you consider the protein contribution of cereals, assuming 70% carbohydrates and 30% proteins? If so, I am satisfied with the model. However, I believe that, like me, other researchers would appreciate having this information available. This isn't just for the purpose of assessing your work but also for the benefit of learning and improving their own methodologies and interpretations, especially considering that CSIA-AA is still cutting-edge.

In our MixSIAR model, we assumed that prehistoric Korean people consumed five different foodstuff groups: 

1. C3 plants (rice, barley, wheat, oat, soybean, Azuki bean) 

2. C4 plants (sorghum, common millet, fox millet) 

3. wild birds (pheasant, wild goose) 

4. terrestrial herbivores

5. marine fish. 

There is no pre-assumption of 70% of carbohydrates and 30% proteins in MixSIAR model. Instead, in this model, we need to consider the diet –tissue offset (TDF) from plants (C3 plant: Δ13C = 5.2‰ and Δ15N = 3.8‰; C4 plant: Δ13C = 4.5‰) to humans and consider the diet –tissue offset (TDF) animals (Δ13C = 1‰ and Δ15N = 3.8‰ to humans respectively.

Furthermore, the MixSIAR output reveals a relative estimated range of five food groups for each individual (e.g., game birds ranging from 18% to 55%). It illustrates the relative spectrum of potential contributions from each food group to the overall diet, summing up to 100%. While the inclusion of additional food groups could alter the proportions of food sources, our selections are grounded in existing knowledge derived from both previous isotopic studies and evidence from food sources in archaeological contexts, including archaeobotany and zooarchaeology.

Reviewer Comments: #Reviewer 2

Reviewer #2: This research paper presents bulk collagen and amino acid stable isotope data (carbon and nitrogen) from bone material (human and animal) of four sites in South Korea. The four sites are Prehistoric cemeteries and represents different Chrono-cultural periods. Four human bones and 4 animal bones are spread on three sites of the Mumum period (Bonze Age culture for this geographical area) and 28 human bones along with 11 animal remains from different species (diversified and representing the resources potentially consumed) are from a site with 1600 burials of the Proto-Three kingdom period (Imdang, more recent than Mumum). The archaeological material is enhanced with modern local domesticated plants (n=9, C3 and C4 cereals) to create a local isotopic baseline. Bulk and CSIA-AA data are discussed thanks to statistical analyses (PCA) and Bayesian models. 

Results indicate (1) different dietary interpretations when comparing bulk and AA data and (2) show how isotopic data improve the knowledge about Prehistoric Korea subsistence strategies, still underdeveloped in the scientific community. The paper deserves publication for both aspects of methodological considerations and archaeological information provided.

Thank you so much for your comments. Here in this paper, we are trying to focus on both methodological progress in terms of amino acid modeling of dietary patterns and the archaeological meaning or significance of this study. This is a balance and we believed we have done this successfully here. 

Specific comments are proposed below.

P 3, 72 and 73: provide the trophic step for both Glu and Phe. 

We corrected this and added this sentence: �15NAA values (�15NGlx-Phe) in bone collagen are useful for estimating the trophic level of humans and animals in food webs.

P4, 102-104: late in the manuscript one can understand the real interest to use isotopic analysis to study diet, but this sentence deserves more information about the archaeological issue in this specific context. 

The use of bulk stable isotope analysis to study human diets are limited and we are trying to use a new technique (compound specific stable isotope analysis) to reconstruct human diets. This is well discussed in the Introduction. 

P4, 113: did you correct the C isotopic data (Suess effect)? if yes mention it in the paper, if not explain why.

There are several reasons why we did not correct for the Suess Effect on the cereal samples.

First, we know of no previous studies on the Suess Effect and how to correct for each amino acid in cereals. It is generally known as 1‰ for bulk carbon isotopes of modern samples, but there are no previous studies of Suess Effects on AAs and so we are not confident to add 1‰ to all of the amino acids in cereals without knowing the real variation in the data of each AA. Thus, we think it is better to keep these measured carbon isotope values of each amino acid as is until this is better determined in the future. 

In addition, even if a 1‰ Suess Effect correction is applied to our results for each amino acid this would not alter the interpretations we have here because the C3 and C4 plants mentioned here have large difference (~10‰) in carbon isotope values of AAs, in comparison to the isotopic difference in the Suess effect (~1‰).

P5 114-116: better explain the archaeological issue of this study.

In response we added the following: Overall, the focus of this study is to reveal the significance of the domesticated grains (rice vs. millet) in Mumun diets, and to more accurately distinguish the dietary sources of the two above-mentioned prehistoric Korean populations using the CSIA-AA (both �13CAA and �15NAA), and to quantify the dietary variability in more detail, using MixSIAR.

P5: Mumum material: explain potential difficulty to get material and the exact anatomical part sampled. Why only 4 remains on 3 sites? how it can reflect a pattern or if not explain better how it improves the previous knowledge (to add also in the conclusion part). The same for Imdang. Explain it is coming from scarified humans and how they can reflect a pattern and if not how it improves the previous knowledge (to add also in the conclusion part). 134-136: push forward this issue and also highlight it in the conclusion.

This work represents a significant advance upon the previous knowledge in Korean Archaeology as it utilizes amino acid dietary models to better understand the dietary habits of the individuals at these two sites. This has been well understood by the Reviewer #1 and we do not feel we need to elaborate this point further in the manuscript. 

We have added the anatomical part of each bone that we sampled in Supplementary Table 1. 

In Korean archaeology, it is extremely difficult to get human and animal bone samples from the Mumun Period. During the Mumun period there was an emergence of a new burial system such as Dolmen and stone-cist burials and these new burials were mainly constructed in the inland mountain areas that are not good for bone preservation due to acidic soil conditions. That is why there is a limited sample size for the Mumun humans. 

Thus, in response we added the following:

It should be noted that it is extremely difficult to obtain humans and animals from the Mumun period as these remains are recovered from inland mountain areas of South Korea that are not good for bone preservation due to acidic soil conditions. Thus, this is the reason why so few specimens from the Mumun sites were analyzed in this study.

P8-9: all the technical part should go in a Supplementary Material file.

We completely disagree. We need to keep this “technical part” in main text because this is very important to the methodological advance of this work. See Reviewer #1. 

P20 conclusion: add sentences about the initial archaeological issues (cf. previous comments) and more explain the archaeological issues about agriculture and game in these two different periods. Explain why these two periods are interesting to compare and explain how despite the very small sample size and the specific origin of the material, the significance of the isotopic data (both bulk collagen and AA).

Previous studies on these two time periods have explained the significance of bulk isotope data [36-37]. In this study, we try to focus on methodological improvement and dietary reconstruction using compound-specific stable isotope analysis and this is clearly stated in the introduction and the conclusions. 

However, we added this sentence for clarification: CSIA-AA provided new information about Prehistoric Korean subsistence strategies, the spread of early agriculture based on the domesticated grains and the subsistence economy of ancient states from the two different-era sites such as Mumun and Imdang.

---

## [Decision Letter · Decision Letter 1]

21 Feb 2024

Enhanced dietary reconstruction of Korean prehistoric populations by combining d 13 C and d 15 N amino acids of bone collagen

PONE-D-23-38093R1

Dear Dr. Choy,

We’re pleased to inform you that your manuscript has been judged scientifically suitable for publication and will be formally accepted for publication once it meets all outstanding technical requirements.

Kind regards,

John P. Hart, Ph.D.

Academic Editor

PLOS ONE

Additional Editor Comments (optional):

Reviewers' comments:

Reviewer's Responses to Questions

**Comments to the Author**

1. If the authors have adequately addressed your comments raised in a previous round of review and you feel that this manuscript is now acceptable for publication, you may indicate that here to bypass the “Comments to the Author” section, enter your conflict of interest statement in the “Confidential to Editor” section, and submit your "Accept" recommendation.

Reviewer #1: All comments have been addressed

Reviewer #2: All comments have been addressed

2. Is the manuscript technically sound, and do the data support the conclusions?

Reviewer #1: Yes

Reviewer #2: Yes

3. Has the statistical analysis been performed appropriately and rigorously? 

Reviewer #1: Yes

Reviewer #2: Yes

4. Have the authors made all data underlying the findings in their manuscript fully available?

Reviewer #1: Yes

Reviewer #2: Yes

5. Is the manuscript presented in an intelligible fashion and written in standard English?

Reviewer #1: Yes

Reviewer #2: Yes

6. Review Comments to the Author

Reviewer #1: Dear authors,

Thank you for the good work in addressing my comments. In my opinion, it is a fantastic piece of work worthy of publication, and this has been my suggestion to the journal.

Good luck in your future endeavours.

Kind regards,

Reviewer 1

Reviewer #2: (No Response)

7. PLOS authors have the option to publish the peer review history of their article (what does this mean?). If published, this will include your full peer review and any attached files.

Reviewer #1: **Yes: **Maria Fontanals-Coll

Reviewer #2: No

---

## [Editor Report · Acceptance letter]

3 Mar 2024

PONE-D-23-38093R1 

PLOS ONE

Dear Dr. Choy, 

I'm pleased to inform you that your manuscript has been deemed suitable for publication in PLOS ONE. Congratulations! Your manuscript is now being handed over to our production team.

Kind regards, 

on behalf of

Dr. John P. Hart 

Academic Editor

PLOS ONE